# Smart Nanocarriers in Cosmeceuticals Through Advanced Delivery Systems

**DOI:** 10.3390/biomimetics10040217

**Published:** 2025-04-02

**Authors:** Jinku Kim

**Affiliations:** Department of Biological and Chemical Engineering, Hongik University, Sejong 30016, Republic of Korea; jinkukim@hongik.ac.kr; Tel.: +82-44-860-2798; Fax: +82-44-866-6740

**Keywords:** nanomaterials, cosmeceuticals, stimuli responsiveness, smart nanocarriers

## Abstract

Nanomaterials have revolutionized various biological applications, including cosmeceuticals, enabling the development of smart nanocarriers for enhanced skin delivery. This review focuses on the role of nanotechnologies in skincare and treatments, providing a concise overview of smart nanocarriers, including thermo-, pH-, and multi-stimuli-sensitive systems, focusing on their design, fabrication, and applications in cosmeceuticals. These nanocarriers offer controlled release of active ingredients, addressing challenges like poor skin penetration and ingredient instability. This work discusses the unique properties and advantages of various nanocarrier types, highlighting their potential in addressing diverse skin concerns. Furthermore, we address the critical aspect of biocompatibility, examining potential health risks associated with nanomaterials. Finally, this review highlights current challenges, including the precise control of drug release, scalability, and the transition from in vitro to in vivo applications. We also discuss future perspectives such as the integration of digital technologies and artificial intelligence for personalized skincare to further advance the technology of smart nanocarriers in cosmeceuticals.

## 1. Introduction

Cosmetics, designed for external skin applications to enhance appearance and provide protection, have encountered challenges such as poor skin retention, limited penetration, and ingredient instability [1,2,3]. To address these issues, advanced skin delivery systems have emerged, enabling controlled and targeted delivery of active ingredients [4,5,6]. Notably, the introduction of liposome-based anti-aging lotions by Christian Dior in the late 1980s marked the beginning of nanoparticle exploration in cosmetics [1,4,7,8]. Nanoparticles, with their enhanced surface area-to-volume ratio and nanoscale size, offer superior skin penetration and improved product quality compared to larger particles, opening new avenues for enhancing the efficacy, safety, and esthetic appeal of cosmeceuticals [9,10,11].

The application of nanotechnology, which is defined by ASTM E56 as technologies that manipulate or incorporate materials with at least one dimension between 1 and 100 nanometers, has revolutionized cosmeceutical development (Figure 1) [12,13]. Nanotechnology enables precise and controlled drug release from nanocarriers, improving stability and facilitating targeted delivery based on interactions among components, drug formulation, and the carrier matrix [14,15,16]. Consequently, the cosmeceutical industry has experienced rapid expansion, driven by significant advancements in nanotechnology that contribute to innovative skincare and cosmetic products [17,18].

The early 2000s witnessed a surge in interest in “nanocosmetics”, the application of nanotechnology to cosmetics [19]. This burgeoning field attracted major cosmetic companies and smaller firms, accelerating research and product development. While the term “nanocosmetics” is now commonplace, its precise definition and the associated benefits and drawbacks have remained a topic of ongoing discussion [20]. Even today, comprehensively explaining the scope and advancement of nanocosmetics poses a challenge [21]. Meanwhile, cosmeceuticals are becoming increasingly popular to meet consumers’ demands for enhancing the appearance and health of skin [22]. Cosmeceuticals are cosmetic products that contain biologically active ingredients that are intended to provide medical or therapeutic benefits [23]. They contain ingredients that are purported to have effects beyond simple cosmetic enhancement, such as anti-aging, skin repair, or protection from environmental damage [24]. The active ingredients in cosmeceuticals include vitamins, antioxidants, peptides, and other bioactive compounds [25]. Consequently, the concept of nanocosmeceuticals revolves around the application of nanotechnology to enhance the effectiveness of cosmetic and cosmeceutical products over the past decade (Figure 2a) [26].

**Figure 1 biomimetics-10-00217-f001:**
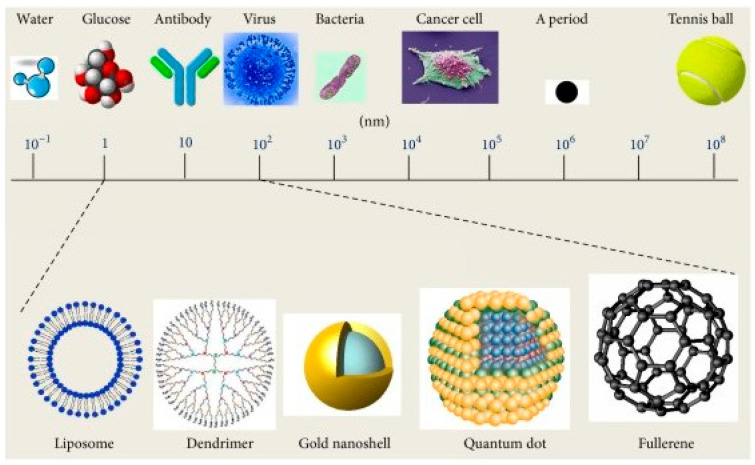
A schematic diagram of different nanoscale materials ranging between 1~100 nm. Organic nanoparticles (polymers, dendrimers); inorganic nanoparticles (calcium phosphate, gold nanoparticles); organic/inorganic hybrids (functionalized gold nanoparticles, nanocomposites); carbon-based (functionalized fullerenes); liposomes and biological nanoparticles (protein and nucleic acid based). Reproduced from [27] under a creative common attribution 4.0 (https://creativecommons.org/licenses/by/4.0/, accessed on 7 March 2025).

Today, the field has progressed significantly, with smart nanocarriers designed to respond to deliver active ingredients to specific skin layers in a controlled manner by sensing internal and external stimuli, such as pH, temperature, or light, ensuring on-demand release of the agents. This responsiveness is crucial for addressing diverse skin concerns, from aging and hyperpigmentation to acne and UV protection [21,28,29,30]. This targeted delivery enhances the efficacy of skincare products like anti-aging creams, sunscreens, and acne treatments. Common examples include lipid nanoparticles, polymeric nanoparticles, and nanocapsules, which encapsulate active ingredients for release based on specific skin conditions [31,32,33].

This review aims to provide a focused overview of the current state of smart nanocarriers used in cosmeceuticals, focusing on their design, fabrication, and application. We will delve into the various types of smart, stimuli-responsive nanocarriers, including thermos, pH-sensitive, and multi-stimuli-responsive nanocarriers, highlighting their unique properties and advantages. Furthermore, we will discuss the biocompatibility of these advanced delivery systems in terms of their potential health risks. Finally, current challenges and future perspectives are also presented. By synthesizing current research and identifying key trends and challenges, this review seeks to provide valuable insights for researchers, industry professionals, and consumers interested in the future of smart nanocarriers in cosmeceuticals.

**Figure 2 biomimetics-10-00217-f002:**
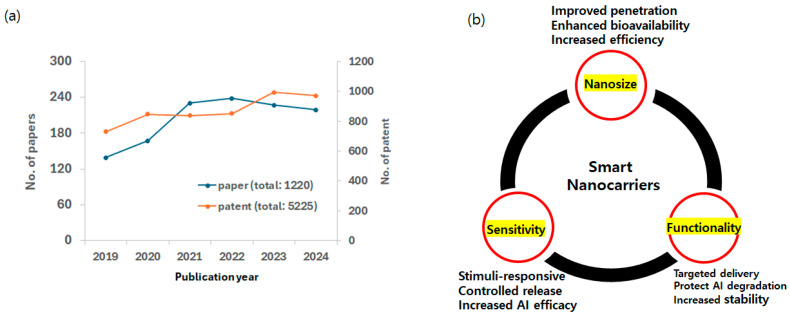
(**a**) Research growth in field of nanocarriers or cosmeceuticals. Paper publication data from Web of Science using “nanocarriers or cosmeceuticals” and patent publication data from USPTO (United States Patent and Trademark Office). (**b**) Integration of benefits of smart nanocarriers in cosmeceuticals.

## 2. Smart Nanomaterials

Smart nanomaterials such as stimuli-sensitive nanocarriers represent a cutting-edge approach to biological applications, leveraging the unique properties of nanoscale materials to enhance product efficacy and delivery. The benefits of small size, sensitivity, and functionality of smart nanocarriers may integrate together and enhance their ability to interact with the microenvironment biologically to maximize the desired performance in cosmeceuticals (Figure 2b) [34]. Cosmeceuticals use smart nanomaterials that respond to external stimuli such as temperature and/or pH. These stimuli-responsive materials are crucial for achieving the desired controlled release and functionality of cosmeceuticals. Nanomaterials, whether natural or synthetic, are designed to interact with living tissues such as skin and must be non-toxic and biocompatible. Consequently, a diverse range of smart biomaterials has been investigated as promising candidates for cosmeceuticals, each exhibiting unique properties and responsiveness to specific stimuli and cosmeceutical applications (Table 1).

### 2.1. Thermosensitive Nanocarriers

Thermosensitive drug delivery is one of the most extensively studied stimuli-responsive approaches and has been widely investigated as a transdermal drug delivery system (TDDS) [35,36,37]. Among these are nanocarrier-based transdermal delivery systems such as liposomes, solid lipid nanoparticles (SLNs), or polymeric nanogels or nanoparticles (usually poly(N-isopropyl acrylamide), PNIPAM) that exhibit a lower critical solution temperature (LCST) [38,39]. These materials exhibit a volume phase transition at a certain temperature (VPTT); a unique, reversible volume change in water near its LCST (32–35 °C), which is driven by a coil-to-globule transition in the polymer network strands [40,41]. Below the LCST, the polymers become hydrophilic and adopt an extended, coiled structure, leading to swelling and a change in shape, whereas they become water insoluble, resulting in gel formation [31,42]. Many studies have utilized this thermoresponsiveness for the controlled release of loaded active ingredients from nanocarriers on topical applications when the surrounding temperature shifts [43]. Among those nanocarriers, thermoresponsive lipid-based nanoparticles have been studied for dermal and topical applications since they allow for the sustained delivery of encapsulated active ingredients to the skin as the temperature varies depending on skin depth [44,45]. For example, Kang et al. developed thermosensitive SLNs for efficient delivery and improved dermal distribution of encapsulated active ingredients, which are difficult to permeate and poorly water soluble, into deep skin layers. They demonstrated that thermosensitive SLNs are excellent topical drug delivery systems, as confirmed by ex vivo and in vivo experiments [46]. Therefore, thermosensitive lipid-based nanoparticles represent a promising and innovative drug delivery system for dermal applications.

**Table 1 biomimetics-10-00217-t001:** Smart nanocarriers used in various cosmeceutical applications.

Stimuli	Mechanism (Mode of Action)	Nanocarrier	Applications	Reference
Thermo	Structural change	SLNs Poloxamers (or Pluronics) nanogels Polyglycerol-based nanogels Porous silica nanogels Polyether-based nanogels Medecassoside liposomes	Dermal delivery TDDS for skin infection treatment Topical drug delivery carrier Skin penetration enhancer Dermal delivery of biomolecules Treatment of inflammatory skin Topical drug delivery Wound healing	[36,38,46,47,48,49,50,51,52,53,54]
pH	Electrostatic repulsion	PLGA-based nanoparticles Eudragit E100 Cellulose phthalates NPs Eudragit L100	Treatment of atopic dermatitis Skin care Dermal carriers Dermal carriers	[32,55,56]
Redox	Cleavage of specific bonds	PEG-block-PLA nanoparticles	Topical application of retinol	[57]
Enzyme	Cleavage of specific peptide sequences	PCL nanofiber patch	Wound healing	[58]
Electro	Structural change	Carbon nanotubes	Transdermal delivery system	[59]
Multi-stimuli	Temp/pH Redox/pH	PNIPAM-co-AAc nanogels PNIPAM-based microgels Liposome modified with acrylic polymers Liposomal sludge	Topical caffeine delivery Dermal delivery of biomolecules Delivery of cosmetic agents Transdermal delivery	[60,61,62,63]

PNIPAM is an increasingly popular choice as a thermosensitive polymeric building block in a nanocarrier for the transdermal delivery of bioactive ingredients. For example, Osorio-Blanco et al. developed thermosensitive polymeric nanocapsules (NCs) around silica nanoparticles (NPs), resulting in the formation of SiO2@NGs as a skin penetration enhancer for skin hydration [50]. N-Isopropylacryl amide (NIPAM) in combination with N-isopropylmethacryl amide (NIPMAM) with different ratios served as thermoresponsive building blocks with dendritic polyglycerol (dPG) as a crosslinker (Figure 3a). These NCs revealed a volume phase transition temperature (VPTT) around 40 °C, enabling controlled drug release at higher temperatures. The images of the skin sections showed that the samples treated with nanocarriers exhibited a higher fluorescence signal than the samples treated with the aqueous dye solution or treated with a nanocarrier without NPs (Figure 3b). A similar strategy was adopted for better transdermal delivery of nanogels fabricated with oligo ethylene glycol (OEG) and dPG as a thermoresponsive polymeric building block and a macro-crosslinker, respectively [38].

In addition, a recent study developed thermoresponsive nanocarriers using mesoporous silica nanoparticles (MSNs) coated with PNIPAM for controlled quercetin delivery to the skin and they showed promising outcomes of the system as an efficient approach for the controlled delivery of antioxidants using a thermal sensitive nanocarrier [51].

Although PNIPAM is the preferred polymer building block for thermosensitive nanocarriers, other polymeric networks have been explored, which include poly(ethylene glycol) methacrylate (PEGDMA) [38], polygycerol derivatives [64], and poly(N-viylcaprolactam) (PVCL) [65]. The nanocarriers fabricated with these polymeric building blocks exhibited the potential for advanced dermal delivery of bioactive ingredients. For example, Calderon and colleagues developed precisely engineered, highly biocompatible, thermosensitive nanogels (tNGs) using oligo ethylene glycol (OEG) as a thermosensitive component and dendritic polyglycerol (dPG) as a crosslinker. The size and volume phase transition temperature (VPTT) of these tNGs can be carefully controlled by surfactant concentration, crosslinker acrylation degree and feed, and OEGMA feed ratio. Preliminary uptake studies of Rhd labeled NGs into human skin demonstrated temperature-dependent internalization of these systems and better penetration in the epidermis than non-thermosensitive counterparts [38]. Furthermore, triblock copolymers poly(ethylene oxide)-b-poly(propylene oxide)-b-poly(ethylene oxide)(PEO-PPO-PEO), known as Poloxamers or Pluronics, have been extensively used for constructing thermosensitive nanocarriers for the delivery of cosmeceuticals due to their excellent biocompatibility approved by US FDA for certain biomedical applications [64,66].

### 2.2. pH-Sensitive Nanocarriers

The natural pH of the healthy skin ranges from pH 4 to 6 depending on the conditions of body and environment, and the maintenance of low acidity of the skin is essential for important skin functions such as homeostasis or the integrity and cohesion of the stratum corneum (SC) [67]. Inflammatory skin conditions often exhibit different pH levels compared to healthy skin [68]. An elevated skin pH compromises the skin’s protective barrier and microbiome, increasing the risk of infection and inflammation, which results in several skin disorders, including atopic dermatitis [67] and acne [68]. The pH-sensitive carriers can be designed to release their therapeutic payload specifically in these altered pH environments, maximizing drug efficacy at the target site [69,70]. Consequently, researchers have been increasingly exploring pH-sensitive nanocarriers for skin treatment for targeted and controlled drug delivery of active ingredients in skin treatments due to such key advantages [71,72,73,74]. For instance, a research group reported an excellent delivery system containing pH-responsive gold nanoparticle-stabilized liposomes for topical antimicrobial delivery. As the delivery system effectively enabled the controlled release of nanoparticle-stabilized liposomes into the bacterial culture, leading to pH dependent fusion with the bacterial membrane, they demonstrated the feasibility of the pH-sensitive nanoparticles as an emerging drug delivery platform for topical applications [75].

pH-sensitive carriers to deliver hydrocortisone-loaded microparticles in a controlled manner to the pH-responsive carriers mainly rely on functional groups that either accept or lose a proton depending on the surrounding acidity [33,76]. These polymers are categorized as either anionic or cationic. A recent study investigated the feasibility of specific pH-sensitive nanoparticles for a transdermal targeted delivery of biomolecules to the affected skin [77]. They used poly(methacrylic acid-co-methyl methacrylate, 1:1), also known as Eudragit L 100, as a polymeric building block to produce a pH-sensitive nanocarrier and demonstrated the enhanced cutaneous penetration of dexamethasone into the skin, analyzed by electron paramagnetic resonance (EPR) and confocal laser scanning microscopy (CLSM) (Figure 4). Similarly, another study used the same atopic dermatitis skin where the pH is elevated, compared to normal skin pH (5.0~5.5). The data revealed that the 10% particle incorporation showed a 26-fold increase in drug release at pH 7 compared to pH 5. In addition, they also demonstrated that the incorporation of these pH-sensitive microparticles into Carbopol and HPMC-based gel formulations showed four-fold greater permeation of the agent into porcine skin after 24 h at pH 7 compared to pH 5 [78].

The pH-sensitive nanocarriers can also be designed to respond to subtle pH changes, allowing precise control over the timing and rate of active agents to the site, which is particularly valuable for chronic skin conditions that require sustained drug delivery [19,79]. For instance, Jung et al. developed pH-sensitive ceramide imbedded PLGA nanocarriers with chitosan coating (Chi-PLGA/Cer) to overcome the limitation of the hydrophobic nature of ceramide and side effects of excessive treatment of skin conditions such as atopic dermatitis (AD). The nanocarrier systems were able to enhance initial adherence to the skin and prevent the initial burst release of ceramide and were degraded by the weakly acidic skin, resulting in the controlled release of ceramide (Figure 5) [55]. It is important to recognize that the benefits of pH-sensitive nanocarriers in skincare often intertwine. Improved drug stability and enhanced penetration frequently occur together [80].

### 2.3. Other Stimuli-Sensitive Nanocarriers

Besides thermosensitive and pH-sensitive nanocarriers, other stimuli-sensitive nanocarriers have been explored for transdermal delivery systems. For example, a redox-responsive poly(ethylene glycol)-block-poly(lactide) (PEG-block-PLA) polymeric nanocarrier containing a disulfide bone was developed to deliver an anti-aging agent (retinol). The authors demonstrated the redox-sensitive behavior of the nanocarriers in the presence of glutathione, susceptible to breaking the disulfide bone of the nanocarriers [57]. In addition, enzyme-sensitive nanocarriers can be designed for transdermal delivery systems since the skin is known to have high enzyme activity, which can be utilized when designing delivery of active ingredients under the biological environment [81]. As an example, Kim et al. designed a nanocarrier by conjugating a genetically engineered epidermal growth factor (EGF) containing matrix metalloproteinase (MMP) cleavage site onto a nonwoven poly(ε-caprolactone)(PCL) fiber mat to release EGF only in the presence of the enzyme. They showed that the enzyme-sensitive nanofibers significantly increased migration and proliferation of human keratinocytes in the presence of MMP-9 compared to the control [58]. An electro-sensitive nanocarrier can also be prepared for the controlled release of active ingredients as a transdermal drug delivery system. For this purpose, Im et al. developed an electro-sensitive nanocarrier fabricated by a semi-interpenetrating polymer network (IPN) containing multi-walled carbon nanotubes responsible for electro-sensitivity. The drug release was observed to increase proportionally with the applied electric voltage, attributed to the voltage-induced dissolution of polyethylene oxide within the semi-IPN [59].

### 2.4. Multiple Stimuli-Sensitive Systems

For enhanced efficiency of smart nanocarriers, one may consider combining multiple stimuli-responsive properties of nanocarriers for the controlled transdermal delivery of active cosmeceutical ingredients. The materials can be designed to respond to multiple stimuli such as temperature, pH, and redox potential for improved specificity and more precisely controlled delivery of bioactive agents [82]. For example, Yamazaki and colleagues developed dual stimuli-responsive liposomes modified with pH- and temperature-sensitive polymers for controlled transdermal delivery (Figure 6) [62]. They were able to show pH and temperature dependence of the release of calcein (model bioactive ingredient) from the smart liposomes modified with the polymers, whereas no content was released from unmodified liposomes at any pH region or at any tested temperatures. Therefore, these smart NPs may have the potential usefulness as a better delivery system for cosmeceuticals or transdermal therapeutics.

Furthermore, temperature-sensitive PNIPAM can be combined with pH-sensitive monomers such as methacrylic acid (MAA), acrylic acid (AAc), or hyaluronic acid (HA) for the development of dual stimuli-responsive transdermal drug delivery systems [61,71]. For this purpose, Abu Samah and Heard developed temperature- and pH-sensitive polyNIPAM copolymerized with AAc termed poly(NIPAM-co-AAc) nanogels to enhance the transdermal delivery of caffeine. They revealed that the permeation data of caffeine-loaded poly(NIPAM-co-AAc) demonstrated the enhanced delivery of the loaded caffeine across the epidermis in comparison to the saturated solution of caffeine by 3.5 orders of magnitude [60].

In certain cases, pH and redox sensitivity can be used simultaneously since a pH gradient and oxidative environment coexist in certain pathological skin conditions. For example, a recent study produced a redox/pH-sensitive nanocarrier, engineered by Eudragit E100-cystamine (EuE100-cyst) and phospholipids for transdermal drug delivery systems, which is triggered by glutathione (GSH) and low pH [63]. The results demonstrated an effective transdermal therapeutic efficacy for the controlled release of corticosteroid through a pig skin model. However, the fabrication of multi-stimuli-responsive systems often requires very complex processes; thus, cost-effective ways for large scale production must be developed before commercial realization.

## 3. Biocompatibility of Nanocarriers

While nanocarriers offer significant benefits in cosmetics, their safety concerns cannot be overlooked. Rigorous research, regulatory oversight, and transparent communication are essential to ensure the safe use of nanocarriers in cosmetic products. Concerns about potential health risks from nanocarriers remain significant, primarily due to the limited availability of long-term toxicological data and the presence of contradictory research results [83,84]. Balancing innovation with safety will be key to the sustainable growth of nanotechnology in the cosmetics industry [85].

### 3.1. Nanomaterials

Currently, there is no globally accepted consensus that identifies nanomaterials as cosmeceutical ingredients. In the US, the FDA has not yet defined nanomaterials in terms of regulatory perspective and has stated that “the current framework for safety assessment is sufficiently robust and flexible to be appropriate for a variety of materials, including nanomaterials” [86]. However, scientists implicitly defined nanomaterials as a term referring to a material or final product designed to have at least one dimension between approximately 1 and 100 nanometers (Figure 1) [15,21]. This is based on the definition given by some important organizations, such as the International American Society for Testing and Materials (ASTM), which is recognized worldwide for the development of international standards. In addition, the ASTM published the first formalized definition of nanotechnology: “any technology that measures, manipulates or incorporates materials and/or resources from 1 to 100 nm”. This concept is very similar to the National Nanotechnology Initiative (NNI)’s definition: “nanotechnology is the development, understanding and control of materials at the nanoscale, ranging from 1 to 100 nm” [87]. Moreover, the FDA published three comprehensive guidance documents concerning the safety issues of nanotechnology, with two of them being related to cosmetics [88]. Based on their recommendations, the FDA-regulated products, including cosmetics, involve the application of nanotechnology, which concerns both the size of the particles and the properties/phenomena depending on size: they will ask (1) “if a material or final product is designed to have at least one external dimension, or internal or surface structure, in the nanoscale range (approximately 1 nm to 100 nm)”; and (2) “if a material or final product is designed to exhibit properties or phenomena, including physical or chemical properties or biological effects, which are attributable to its size, even if these dimensions are outside the nanoscale range, down to one micrometer (1000 nm)” [89].

Although nanomaterials are a new class of biomaterials and offer opportunities for better cosmeceutical functions, they are subject to thorough screening to ensure the safety of consumer health [90]. The production and use of nanomaterials may result in unknown health risks since the exposures of biological systems to nanomaterials of this size have not been adequately studied [85]. Furthermore, a nanomaterial may have different biological interactions than the same material in larger dimensions [91]. Properties of nanomaterials such as small size, large surface area, and high reactivity that make them unique and impart tremendous potential for technological advances are also the very properties that may be responsible for adverse effects [92,93]. For example, the small size of nanoparticles allows them to penetrate deeper layers of the skin, potentially reaching viable cells and even the bloodstream, which raises concerns about the potential for systemic exposure and accumulation of nanomaterials in the body [94]. How nanomaterials enter the body is critical to assessing their safety. While cutaneous exposure is the primary route for cosmetics, it is unclear if the nanomaterials penetrate through the stratum corneum, which is the outermost layer of the epidermis [95]. Skin conditions (e.g., damaged skin, eczema, psoriasis) can significantly influence nanoparticle penetration, resulting in increasing health risk. In addition, special attention should also be paid to exposures by inhalation as well as ingestion of nanomaterials [96].

### 3.2. Regulatory Aspects

Since the Europe (EU) and the US are the two major markets for cosmetic and cosmeceutical products, Table 2 highlights the regulatory oversight for nanocarriers in cosmetics and cosmeceuticals in the EU and the US. Currently, the EU has stricter regulations on nanomaterials or nanocarriers used in cosmetic products compared with the US [17].

### 3.3. Biocompatibility Testing

#### 3.3.1. In Vitro Testing

Due to the unique properties of nanomaterials in cosmetics and cosmeceuticals, which drive product function, consumers may face potential health risks. Therefore, rigorous safety reviews are essential for each nanomaterial, which must include testing on nano-specific characteristics, such as skin penetration and inhalation risks [97]. Most regulatory agencies such as the FDA require in vitro testing prior to in vivo testing. The FDA safety assessments should consider factors like physicochemical properties, size distribution, shape, solubility, and potential impurities. Furthermore, it is crucial to determine possible exposure routes and gather comprehensive toxicological data, including dermal penetration, inhalation, genotoxicity, and irritation potential [98]. In addition, the ASTM guidelines offer thorough validation of nanomaterials for in vitro cytotoxicity (ASTM E2526) using two methods such as 3-(4,5-Dimethylthiazolyl-2)-2,5-diphenyltetrazolium bromide (MTT) reduction and lactate dehydrogenase (LDH) release. They also include an in vitro inflammation test (ASTM E2525) to determine nanoparticle stimulation on the inhibition of the maturation of certain bone marrow cells, which may be particularly sensitive to nanoscale materials.

#### 3.3.2. In Vivo Testing and Clinical Trial

Obviously, the in vitro testing of smart nanocarriers has limitations in accurately replicating the complex biological responses to stimuli. In addition, the market approval of nanocarriers requires in vivo testing to further determine the biocompatibility with organisms. Despite existing products and patents, thorough in vivo testing is crucial, including evaluations of metabolization, permeation, biodistribution, and elimination of subproducts [19,99]. For example, permeation tests are essential for evaluating the safety of smart transdermal carriers, but regulatory and ethical concerns are reducing or even eliminating the use of animal skin in permeation studies [14]. Therefore, ex vivo testing can be considered for proof-of-concept studies of new smart nanocarriers. In general, human skin biopsies are the gold standard, but their limited availability necessitates the use of defrosted skin although it exhibits a lower barrier function compared to in vivo human skin, which is a significant drawback [100]. Over recent decades, significant efforts have been made to develop artificial membranes and cultured 3D human skin models. Consequently, reconstructed skin models are becoming increasingly promising and play a prominent role in additional biocompatibility testing in the future [101].

Despite extensive research on advanced nanocarriers in cosmetics and cosmeceuticals, there is a significant lack of studies demonstrating how well in vitro and in vivo safety data predict human clinical outcomes. For this reason, human clinical trials are being conducted to verify the safety of nanocarriers as well as the efficacy of cosmeceutical products for improving skin conditions [102]. As a result, it was reported that nanoparticles can cause endocrine disruption and immunological effects [21]. They can also induce toxicity in various organ systems, including pulmonary, musculoskeletal, cardiovascular, neurological, respiratory, reticuloendothelial, and circulatory systems [103]. Therefore, more investigations are needed to ensure the safety of smart nanocarriers prior to market approval.

## 4. Challenges and Future Perspectives

Despite remarkable advancements of smart nanomaterials-based delivery systems in cosmeceuticals, a majority of stimuli-responsive nanocarriers are still in the early stages of development and the optimization of the synthesis procedures is needed before they can be available to consumers. First, it is very difficult to achieve precise control over the timing and amount of active ingredients required for optimal efficacy. Thus, the need for precise control over the “response” to the applied “stimulus” makes their clinical translation challenging [30,104]. In addition, unwanted or premature release of the active ingredients before the stimulus is applied to the carriers is also a major challenge [105]. Furthermore, skin conditions (skin pH, temperature, and moisture levels) can fluctuate, affecting the release of active ingredients, making it challenging to ensure consistent and predictable responses from the nanocarriers [106]. Therefore, designing nanocarriers that release active ingredients at the desired rate and duration remains a significant challenge. Maintaining the stability of active ingredients and stimuli-responsive nanocarriers under various storage conditions is essential since active ingredients and nanocarrier materials can degrade over time, affecting product efficacy and safety [107]. While numerous stimuli-responsive nanosystems have been evaluated in vitro, there is a significant gap between in vitro and in vivo applications, especially for topical or transdermal applications, demanding urgent attention, an aspect that needs immediate focus [108]. The scalability and manufacturing of stimuli-sensitive nanocarriers at a large scale, which can be complex and expensive, require tremendous efforts and financial burden; hence, developing cost-effective and scalable production processes should be essential for commercial viability [109].

While several major challenges are yet to be overcome, the future of smart stimuli-sensitive nanocarriers in cosmeceuticals holds immense potential, driven by ongoing advancements in nanotechnology and a growing demand for personalized and effective skincare solutions. Future research should focus on overcoming current challenges while addressing potential risks to ensure consumer safety and product efficacy. For example, nanocarriers will be integrated with digital technologies, such as wearable sensors and mobile apps, to provide real-time monitoring of skin health and personalized skincare recommendations [110,111]. Artificial intelligence (AI), especially generative AI, will be used to analyze vast amounts of data and develop optimized nanocarrier formulations for specific skin concerns [112,113,114].

## Figures and Tables

**Figure 3 biomimetics-10-00217-f003:**
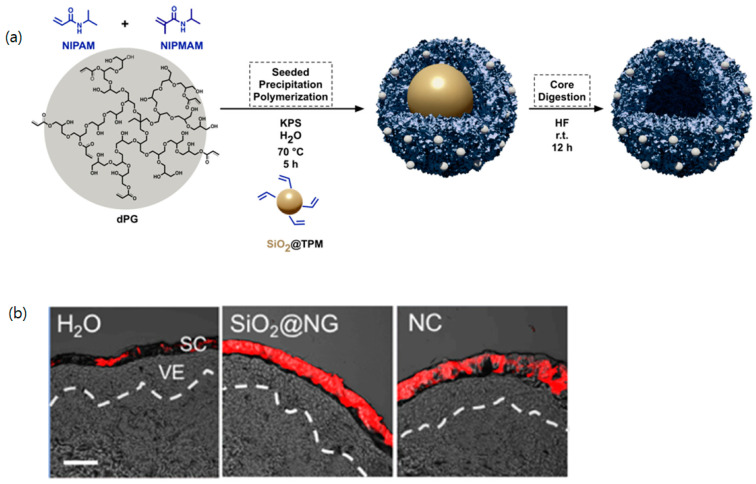
(**a**) Schematic design of thermosensitive nanocarriers using silica nanoparticles; (**b**) representative images of skin section, showing different intensity of penetration dye. Scale bar = 50 μm. Reproduced from [50] under creative common attribution 4.0 (https://creativecommons.org/licenses/by/4.0/, accessed on 7 March 2025).

**Figure 4 biomimetics-10-00217-f004:**
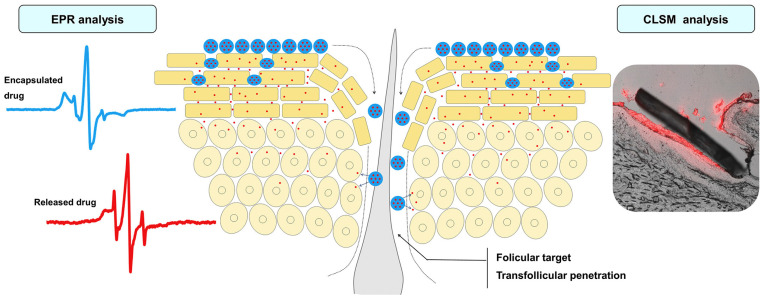
Penetration and release of active ingredients from pH-sensitive nanoparticles. reproduced from [77] under creative common attribution 4.0 (https://creativecommons.org/licenses/by/4.0/, accessed on 7 March 2025).

**Figure 5 biomimetics-10-00217-f005:**
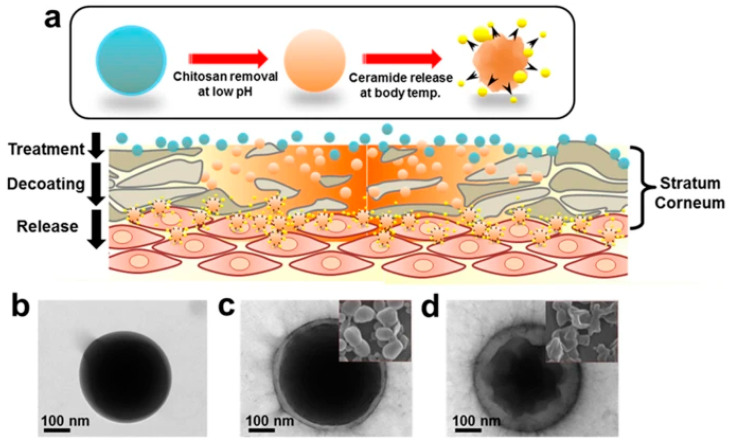
(**a**) Schematics of design of chitosan-PLGA/ceramide treatment on atopic dermatitis (AD) lesion and electron microscopic images of shape of (**b**) PLGA nanoparticles, (**c**) chitosan coating, and (**d**) shrinkage of PLGA. Reproduced from [79] under creative common attribution 4.0 (https://creativecommons.org/licenses/by/4.0/, accessed on 7 March 2025).

**Figure 6 biomimetics-10-00217-f006:**
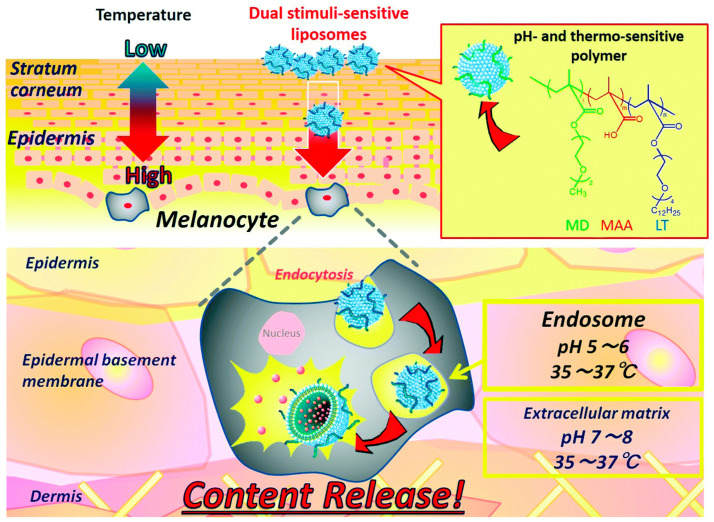
Dual stimuli-responsive liposomes for transdermal drug delivery system, which responds to body temperature at epidermis and acidic pH at endosome. Reproduced from [57] under creative common attribution 4.0 (https://creativecommons.org/licenses/by/4.0/, accessed on 7 March 2025).

**Table 2 biomimetics-10-00217-t002:** EU and US regulatory landscape for nanocarriers in cosmetics and cosmeceuticals.

	EU	US
Regulating organizations	EC, SCCS, EUON	FDA, NTF, NNI, PCPC
Regulatory guideline	Robust regulation (EC No. 1223/2009) for nanomaterials in cosmetics	Lack of specific regulations for nanomaterials in cosmetics
Notification and labeling	Companies must notify the EC about nanomaterials used Products must clearly label nanomaterials with “(nano)” in the ingredient list	The FDA monitors nanotechnology in cosmetics but no notification required Do not require specific “nano” labeling
Safety guidance	The Scientific Committee on Consumer Safety (SCCS) provides guidance on nanomaterial safety assessment, with ongoing updates Each cosmetic brand needs a responsible person and must register products with the Cosmetic Products Notification Portal (CPNP) Toxicological data cannot be obtained through animal testing, requiring alternative methods like in vitro and ex vivo	The FDA provides safety guidance for nanomaterials in cosmetics to help manufacturers identify and evaluate potential safety issues
Definition review	Reviewing the definition of nanomaterials in cosmetics to align it with other recommendations.	No definition review

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
