# Peer review of "Smart Nanocarriers in Cosmeceuticals Through Advanced Delivery Systems"

_biomimetics, 2025, doi:10.3390/biomimetics10040217_

Round 1

Reviewer 1 Report

Comments and Suggestions for Authors
  1. What is the novelty of this review. Similar reviews are available. https://www.mdpi.com/2310-2861/8/3/173  ,https://www.mdpi.com/2079-9284/11/1/20 https://www.taylorfrancis.com/chapters/edit/10.1201/9780429299520-12/smart-delivery-systems-personal-care-cosmetic-products-fanwen-zeng-nilesh-shah https://www.degruyter.com/document/doi/10.1515/rams-2022-0282/html https://www.sciencedirect.com/science/article/abs/pii/S0927776522001230 
  2. The plagiarism is 31%.
  3. The author has not made a single figure by themselves, all the figures are adopted.
  4. The author should add patent table.
  5. The author should add a table illustrating applications of different Smart Nanocarriers in Cosmeceuticals.
  6.  The author should cover all the relevant categories of smart nanocarriers (hybrid or multifunctional nanocarriers) for Cosmeceuticals.
  7. The author should add two in vivo studies for any nanocarrier for Cosmeceuticals with adapted figures.
  8. The author should add the discussion on regulatory aspects( FDA, EMA, or other global regulations) regarding nanomaterial safety in cosmetics?
  9. There is no point no3, from Point no 2 to direct point no 4, please check this and correct.
  10. There is no conclusion written by the author, add it.
  11. They were able to show pH and temperature dependence of the controlled release model active agent (calcein) from the smart liposomes modified with the polymers. How much release were reported, add reported data.
  12. The results demonstrated aneffective transdermal therapeutic efficacy for controlled release of corticosteroid through a pig skin model. how much release were reported. 
  13. While the manuscript is generally well-written, certain sections contain minor grammatical errors and awkward phrasing. A thorough proofreading or language review would improve readability and flow.
  14. Some references appear incomplete or inconsistent in formatting. Please ensure that all citations follow the journal's required format and include complete details such as volume, issue, and DOI where applicable.
  15. List of abbreviations should be added after the conclusion.
  16.  
Comments on the Quality of English Language

Average

Author Response

  1. What is the novelty of this review. Similar reviews are available. https://www.mdpi.com/2310-2861/8/3/173

https://www.mdpi.com/2079-9284/11/1/20

https://www.taylorfrancis.com/chapters/edit/10.1201/9780429299520-12/smart-delivery-systems-personal-care-cosmetic-products-fanwen-zeng-nilesh-shah

https://www.degruyter.com/document/doi/10.1515/rams-2022-0282/html

https://www.sciencedirect.com/science/article/abs/pii/S0927776522001230

Response: Although similar reviews are available in the literature as the reviewer mentioned, the novelty of the current review is to provide a focused and concise overview of smart nanocarriers (stimuli responsive nanocarriers in particular) used in cosmeceuticals. The articles that the reviewer mentioned above are related to the overview of nanotechnologies used in cosmetics and cosmeceuticals or specific smart carriers (e.g., tissue composites) or nanocarriers prepared by the smart encapsulation process technologies. Therefore, the author believes that the current manuscript can be published to provide more focused and concise overview of smart nanocarriers (stimuli-responsive nanocarriers in particular) in cosmeceuticals. The author included the review articles mentioned by the reviewer in the manuscript.

  1. The plagiarism is 31%.

Response: The author revised and refined the manuscript to resolve the plagiarism issue in the manuscript.

  1. The author has not made a single figure by themselves, all the figures are adopted.

Response: The author added a figure to provide research growth in the field of nanocarriers and cosmeceutical, and integration of the benefits of smart nanocarriers used in cosmeceuticals. (Figure 2)

  1. The author should add patent table.

Response: The author included literature growth including patents published in the US. (Figure 2a)

  1. The author should add a table illustrating applications of different Smart Nanocarriers in Cosmeceuticals.

Response: The author added a table to illustrate specific applications of different smart nanocarriers (Table 1)

  1.  The author should cover all the relevant categories of smart nanocarriers (hybrid or multifunctional nanocarriers) for Cosmeceuticals.

Response: The author added a table to cover relevant categories of smart nanocarriers and their applications. (Table 1)

  1. The author should add two in vivo studies for any nanocarrier for Cosmeceuticals with adapted figures.

Response: The author has tried to add in vivo study data as much as possible for any nanocarrier as the reviewer recommended (lines 200~205).

  1. The author should add the discussion on regulatory aspects (FDA, EMA, or other global regulations) regarding nanomaterial safety in cosmetics?

Response: The author summarized the regulatory aspects regarding nanomaterial safety in cosmetics in table 2.

  1. There is no point no3, from Point no 2 to direct point no 4, please check this and correct.

Response: The author corrected the point sequence.

  1. There is no conclusion written by the author, add it.

Response: The author think that concluding remarks have been added in the challenges and future perspectives section with open conclusion.

  1. They were able to show pH and temperature dependence of the controlled release model active agent (calcein) from the smart liposomes modified with the polymers. How much release were reported, add reported data.

Response: The data were included in the manuscript as pointed out (lines 258~59).

  1. The results demonstrated an effective transdermal therapeutic efficacy for controlled release of corticosteroid through a pig skin model. how much release were reported. 

Response: The data were included in the manuscript as suggested (lines 200~205).

  1. While the manuscript is generally well-written, certain sections contain minor grammatical errors and awkward phrasing. A thorough proofreading or language review would improve readability and flow.

Response: The author revised typo, grammatical errors, and awkward phrasing throughout the manuscript to improve readability and flow.

  1. Some references appear incomplete or inconsistent in formatting. Please ensure that all citations follow the journal's required format and include complete details such as volume, issue, and DOI where applicable.

Response: The author corrected the references with complete and consistent in formatting.

  1. List of abbreviations should be added after the conclusion.

Response: List of abbreviations were added after the conclusion as pointed out.

Reviewer 2 Report

Comments and Suggestions for Authors

The paper titled "Smart Nanocarriers in Cosmeceuticals through Advanced Delivery Systems" by Jinku Kim reviews the application of nanotechnologies in cosmeceuticals. It details the advancements in smart nanocarriers designed to enhance skin delivery systems, focusing on various stimuli-responsive nanocarriers including thermo-, pH-, and multi-stimuli sensitive types. The review addresses biocompatibility and potential health risks associated with these nanomaterials, outlining the advantages they provide in cosmeceutical applications, particularly in improving the controlled release and efficacy of skincare treatments. Additionally, it discusses the current challenges and future directions in the integration of digital technologies and artificial intelligence for personalized skincare solutions.

  1. While the manuscript provides a general overview of different types of nanocarriers, more detailed descriptions of the mechanisms by which these nanocarriers enhance skin penetration and stability of active ingredients will be benificial.
  2. The discussion on biocompatibility and safety is crucial but appears somewhat generic. Could the authors expand on specific biocompatibility tests or clinical trials that have been conducted, particularly those that relate directly to the nanocarriers discussed?
  3. The review mentions various advantages of nanocarriers in cosmeceuticals but lacks quantitative data to support these claims. Including data on the increase in skin retention, penetration depth, or improvement in ingredient stability could strengthen the paper’s arguments.
  4. Figure 2B does not have size with scale bar.
  5. The manuscript highlights the use of thermo- and pH-sensitive nanocarriers but could provide more specific examples of how these systems have been optimized for particular cosmeceutical applications. For example, how do changes in skin pH during different dermatological conditions affect the release of active ingredients? Including more detailed scenarios or experimental results would provide clearer insights into the practical applications and effectiveness of these nanocarriers.

Author Response

The paper titled "Smart Nanocarriers in Cosmeceuticals through Advanced Delivery Systems" by Jinku Kim reviews the application of nanotechnologies in cosmeceuticals. It details the advancements in smart nanocarriers designed to enhance skin delivery systems, focusing on various stimuli-responsive nanocarriers including thermo-, pH-, and multi-stimuli sensitive types. The review addresses biocompatibility and potential health risks associated with these nanomaterials, outlining the advantages they provide in cosmeceutical applications, particularly in improving the controlled release and efficacy of skincare treatments. Additionally, it discusses the current challenges and future directions in the integration of digital technologies and artificial intelligence for personalized skincare solutions.

  1. While the manuscript provides a general overview of different types of nanocarriers, more detailed descriptions of the mechanisms by which these nanocarriers enhance skin penetration and stability of active ingredients will be beneficial.

Response: The author included a table to provide detailed mechanisms to enhance skin penetration and stability of active ingredients as recommended (Table 1).

  1. The discussion on biocompatibility and safety is crucial but appears somewhat generic. Could the authors expand on specific biocompatibility tests or clinical trials that have been conducted, particularly those that relate directly to the nanocarriers discussed?

Response: The author included a more detailed in vivo biocompatibility and safety of smart nanocarriers including regulatory aspects (FDA, EMA, etc) (lines 336~387 and table 2).

  1. The review mentions various advantages of nanocarriers in cosmeceuticals but lacks quantitative data to support these claims. Including data on the increase in skin retention, penetration depth, or improvement in ingredient stability could strengthen the paper’s arguments.

Response: The data was included to strengthen the paper's arguments as recommended (lines 200~205, lines 258~259).

  1. Figure 2B does not have size with scale bar.

Response: The author added the size on the scale bar in Figure 3b.

  1. The manuscript highlights the use of thermo- and pH-sensitive nanocarriers but could provide more specific examples of how these systems have been optimized for particular cosmeceutical applications. For example, how do changes in skin pH during different dermatological conditions affect the release of active ingredients? Including more detailed scenarios or experimental results would provide clearer insights into the practical applications and effectiveness of these nanocarriers.

Response: The added a table (table 1) to highlight mechanisms, specific stimuli-responsive nanocarriers and their specific applications.

Round 2

Reviewer 1 Report

Comments and Suggestions for Authors

The author has successfully addressed all the reviewer’s comments and incorporated the suggested revisions. The manuscript now meets the required standards and is suitable for publication.

Reviewer 2 Report

Comments and Suggestions for Authors

Agree to publish